# Evolution and Expression Patterns of the Fructose 1,6-Bisphosptase Gene Family in a Miracle Tree (*Neolamarckia cadamba*)

**DOI:** 10.3390/genes13122349

**Published:** 2022-12-13

**Authors:** Qingmin Que, Xiaohan Liang, Huiyun Song, Chunmei Li, Pei Li, Ruiqi Pian, Xiaoyang Chen, Wei Zhou, Kunxi Ouyang

**Affiliations:** 1College of Forestry and Landscape Architecture, South China Agricultural University, Wushan Road 483, Tianhe District, Guangzhou 510642, China; 2Zhaoqing Branch Center of Guangdong Laboratory for Lingnan Modern Agricultural Science and Technology, Zhaoqing 526238, China; 3Guangdong Key Laboratory for Innovative Development and Utilization of Forest Plant Germplasm, South China Agricultural University, Wushan Road 483, Tianhe District, Guangzhou 510642, China

**Keywords:** phylogenetic analysis, fructose 1,6-bisphosptase, *Neolamarckia cadamba*, bioinformatics analysis

## Abstract

*Neolamarckia cadamba* (*N. cadamba*) is a fast-growing tree species with tremendous economic and ecological value; the study of the key genes regulating photosynthesis and sugar accumulation is very important for the breeding of *N. cadamba*. Fructose 1,6-bisphosptase (*FBP*) gene has been found to play a key role in plant photosynthesis, sugar accumulation and other growth processes. However, no systemic analysis of *FBPs* has been reported in *N. cadamba*. A total of six *FBP* genes were identifed and cloned based on the *N. cadamba* genome, and these *FBP* genes were sorted into four groups. The characteristics of the *NcFBP* gene family were analyzed such as phylogenetic relationships, gene structures, conserved motifs, and expression patterns. A *cis-acting* element related to circadian control was first found in the promoter region of *FBP* gene. Phylogenetic and quantitative real-time PCR analyses showed that *NcFBP5* and *NcFBP6* may be chloroplast type 1 *FBP* and cytoplasmic *FBP*, respectively. FBP proteins from *N. cadamba* and 22 other plant species were used for phylogenetic analyses, indicating that *FBP* family may have expanded during the evolution of algae to mosses and differentiated cpFBPase1 proteins in mosses. This work analyzes the internal relationship between the evolution and expression of the six *NcFBPs*, providing a scientific basis for the evolutionary pattern of plant *FBPs*, and promoting the functional studies of *FBP* genes.

## 1. Introduction

*N. cadamba* (Roxb.) Bosser (Rubiaceae) is mainly distributed in Burma, India, Vietnam, Sri Lanka and other places in Southern Asia, as well as Papua New Guinea, Australia and other countries in the South Pacific [1]. At the World Forestry Congress in 1972, *N. cadamba* was described as ‘a miraculous tree’ due to its fast-growing. Under normal conditions, it can attain a height of 17.67 m and a trunk diameter of 25.3 cm at breast height within nine years [2]. As early as 1933, Indonesia began to establish the plantation of *N. cadamba*, and it has been introduced to African and Central American countries as a major industrial timber species [3]. *N. cadamba* is not only used for pulp and paper production, but also in furniture making [4]. Its flowers, fruit, leaves and bark are widely used in modern and traditional medicine [5]. Therefore, it has potential, in suitable regions, to meet the increasing demand for wood products.

Fructose 1,6-bisphosptase (FBPase, EC 3.1.3.11), first discovered by Gomori in 1943 [6], is widely found in animals and plants. It can catalyze the irreversible hydrolysis of fructose-1,6-bisphosphate to fructose-6-phosphate and inorganic phosphate. Two FBPase isozymes have been identified in eukaryotic cells of photosynthesis, with one in the cytoplasm (cyFBPase), the key enzyme of gluconeogenesis, and the other one in the chloroplast (cpFBPase), the rate-limiting enzyme involved in the Calvin cycle [7]. CpFBPase isozymes are widely distributed in photosynthetic organisms, such as bacteria [8], cyanobacteria [9], green algae [10], lichens [11] and higher plants [12]. 

FBPase in most higher plants exists in the form of monomer, dimer and tetramer, among which tetramer FBPase has catalytic activity [13]. Both chloroplast and cytoplasmic fructose-1,6-diphosphatase can be highly regulated, but the regulatory mechanism is different between them. CyFBPase plays a regulatory role in gluconeogenesis and is strongly inhibited by adenosine monophosphate (AMP) and fructose-1,6-diphosphate; on the contrary, cpFBPase plays a regulatory role in photosynthesis and is not sensitive to AMP and fructose-1,6-diphosphate [14].

Considerable effort has been made to investigate the roles of FBPase on plant growth, biotic stress and abiotic stress. Miyagawa et al. [15] transferred cyanobacteria chloroplast FBPase into tobacco, which increased the photosynthesis and sugar accumulation of transgenic tobacco and accelerated its plant growth. Thorbjornsen et al. [16] successfully isolated potato cpFBPase gene and analyzed its transformation. Compared with wild potato tubers, the transformed plants contained higher chloroplast FBPase activity and accumulated more starch. In *Arabidopsis thaliana*, it was found that the lack of *cFBP1* resulted in lower soluble sugar content, less starch accumulation and higher superoxide dismutase (SOD) activity, in addition to smaller rosette size and lower photosynthetic rate [12]. The mutant also experienced some developmental changes, including stomatal opening defects and an increase in the number of root vascular layers. In addition, cpFBPase also plays an important role under high temperature or drought in *Pyropia haitanensis* [17]. FBPase also plays an important role in biotic stress. The expression of *PmFBP* gene in insect-resistant varieties was significantly higher than that of the control, speculating that *PmFBP* gene played a key role in the defense process of insect resistance of *Pinus massoniana* [18].

As FBPase plays a key role in plant photosynthesis, sugar accumulation and other growth processes, the study of FBPase is important for *N. cadamba* breeding. In the study, the *NcFBP* gene family members were cloned in the *N. cadamba* genome, followed by comprehensive analyses, including sequence alignment, gene structures, phylogenetic analyses, conserved motifs, and conserved domains. Therefore, the expression patterns of *FBP* genes in different tissues in *N. cadamba* and their phylogenetic analysis were investigated.

## 2. Materials and Methods

### 2.1. Identification and Cloning of FBP Genes in N. cadamba and Identification in 22 Other Species

The predicted *NcFBP* genes were identified as follows: first, the *N. cadamba* genomic sequences were downloaded from the National Center for Biotechnology Information (NCBI) BioProject database [19] and BioEdit biological sequence alignment editor software was used to create a local database file. Next, a hidden Markov model (HMM) program was used for gene prediction against the local database through the BlastP method (E value < 10^−10^). *NcFBP* gene sequences for potential protein-coding segments were searched by ‘ORF prediction’ in TBtools [20], and all candidates were examined for the FBP domain in the Pfam databases. Reverse transcription polymerase chain reaction (RT-PCR) was used to amplify full-length ORFs of these *NcFBP* genes. Total RNA from each sample was isolated using CTAB plus the OMEGA Plant RNA isolation kit, as described previously [21]. Total RNA (0.5 µg) was reverse transcribed into the first strand cDNA following the PrimeScript II First Strand cDNA Synthesis Kit (TaKaRa Bio, Tokyo, Japan) instructions. Candidate genes were amplified using KOD FX polymerase (Toyobo Co., Osaka, Japan) with the following thermocycling conditions: 94 °C for 2 min, 35 cycles (94 °C for 30 s, 53 °C for 30 s, and 72 °C for 40 s), with a final extension at 72 °C for 2 min. The amplified products were sequenced by Beijing Genomics Institute (China) and the sequences were used for further investigation.

To investigate variation of FBP gene family members, we used the HMM software package to identify putative FBP proteins in 22 other species′ completely sequenced genomes, including dicot and monocot plants, ferns, moss, and algae, followed by querying the candidates against the Phytozome database (http://www.phytozome.net/, accessed on 27 May 2021) to confirm the presence of the FBP domain. In total, 151 *FBPs* were identified in these 23 genomes (Table 1). The amino acid sequences were analyzed using the web-based Multiple Expectation–Maximization for Motif Elicitation (MEME) program, version 4.11.4, to identify and analyze conserved motifs [22]. The maximum number of motifs was set to 10 and the optimum motif width was set to ≥6 and ≤50. Finally, conserved domains were identified with the web-based Pfam search program [23] (http://pfam.xfam.org/, accessed on 27 May 2021). The intron–exon structures were drawn by the TBtools [20].

### 2.2. Analysis of Cis-Acting Elements of Promoter

The promoter sequences of *NcFBP* gene family members (2.0 kb upstream of transcription initiation site) were obtained from the *N. cadamba* genome, and the *cis-acting* elements of *NcFBP* gene family promoters were predicted by PlantCARE website (http://bioinformatics.psb.ugent.be/webtools/plantcare/html/, accessed on 30 May 2021).

### 2.3. Multiple Sequence Alignment and Phylogenetic Analyses of FBP Family

For phylogenetic analyses among the NcFBPs, the full length protein sequences were multiply aligned by the ClustalX2.1 program with default parameters, followed by constructing a phylogenetic tree with the maximum likelihood (ML) method using the MEGA X program [24]. The bootstrap values were calculated with 1000 replications. These alignments among 23 species were used to generate a phylogenetic tree with the neighbor-joining method using the Poisson correction in 1000 bootstrap test replicates [25].

### 2.4. Preparation of Plant Materials

The sampled *N. cadamba* plants were planted at 120 m northwest of the College of Forestry and Landscape Architecture, South China Agricultural University (113.36 E, 23.16 N) in 2011. The roots, cambium, phloem tissue, bark, young leaves, old leaves, buds and fruit were collected in July 2020, and all samples were immediately frozen in liquid nitrogen and stored at −80 °C in a refrigerator for further study. Each tissue was collected from three individual plants representing three biological replicates. 

### 2.5. Quantitative Reverse Transcription-Polymerase Chain Reaction (qRT-PCR)

The RNA isolation, cDNA synthesis, qRT-PCR and the selection of reference genes followed the methods described previously [21,26]. qRT-PCR amplification was performed on a LC480 instrument (Roche Diagnostics, Basel, Switzerland) with three technical replicates under the conditions as follows: 95 °C for 30 s, followed by 40 cycles (95 °C for 5 s, annealing at 60 °C for 30 s, and 72 °C for 30 s). The expression level of the roots was used as a control, and the relative expression levels were calculated using the 2^−ΔΔCT^ method [27].

## 3. Results

### 3.1. Identification and Physicochemical Properties of NcFBP Family in N. cadamba

The whole genome amino acid sequence of *N. cadamba* was compared with *A. thaliana* FBP amino acid sequence by TBtools, and six genes encoding FBP in *N. cadamba* were screened, named as *NcFBP1*~*NcFBP6,* respectively. The nucleotide arrangement information of *NcFBPs* gene is shown in Table A1. In order to further understand the function of *NcFBPs* in the growth and development of *N. cadamba*, the physical and chemical properties of the protein sequences of the six FBP genes were analyzed (Table 2). The results showed that the length of NcFBP protein sequence ranged from 340 amino acid residues (aa) (NcFBP4) to 409 aa (NcFBP2), with an average length of 386 aa, which was conserved in evolution. The isoelectric points (pIs) of the FBP proteins ranged from 5.25 (NcFBP5) to 6.27 (NcFBP3), and all pIs were less than 7, indicating that all NcFBP were negatively charged and acidic. The molecular weights (MWs) ranged from 37.1957 kDa (NcFBP6) to 44.3803 kDa (NcFBP4). The instability index of NcFBP protein ranges from 35.88 (NcFBP1) to 44.59 (NcFBP3). Among them, the instability index of NcFBP3 and NcFBP4 were more than 40, indicating that they are less stable than the other members. The aliphatic index of NcFBP protein ranges from 84.56 (NcFBP6) to 94.04 (NcFBP2). The higher the aliphatic index, the better the stability of the protein, which is beneficial for the protein to play a role in different environments. The grand average hydrophobicity of all NcFBP proteins was negative (between −0.223 and −0.092), indicating that all members are hydrophilic proteins. 

### 3.2. Bioinformatics Analysis of FBP Family in N. cadamba

In order to explore the evolutionary relationship among members of NcFBP protein family, a phylogenetic tree was constructed by ML maximum likelihood method. The result is shown in Figure 1a. The NcFBPs can be divided into four subfamilies: NcFBP1 as the member of subfamily I, NcFBP2 as the member of subfamily II, NcFBP6 as the member of subfamily III, NcFBP3, and NcFBP4 and NcFBP5 as the members of subfamily IV.

The gene structure analysis of *NcFBPs* family in *N. cadamba* is shown in Figure 1b. The three genes of subfamily IV were similar in structure, while the three genes were also different. *NcFBP4* had no untranslated regions (UTR), *NcFBP3* had a small UTR at the 5′-end and a long UTR at the 3′-end, while *NcFBP5* only had a small UTR at the 3′-end. There was only one gene in subfamily I, subfamily II or subfamily III, and the gene structures were quite different among them. In terms of the number of exons, there were 12 exons in subfamily III, significantly more than that in other three subfamilies containing 3–10 exons, indicating that there were great differences in gene structure among *NcFBPs* subfamilies.

In order to explore the evolutionary diversity of NcFBP proteins, the conserved domains of six NcFBP protein sequences were analyzed by Pfam (Figure 1c). The result showed that all NcFBP proteins have FBPase (PF00316) domain, indicating that the NcFBP proteins were highly conserved. The NcFBP proteins were analyzed by MEME online analysis software, and 10 conserved motifs were obtained as shown in Figure 1d, named motif1~motif10, respectively. All of the NcFBP protein sequences contained motif2, motif3 and motif4, NcFBP1 had the least number of motifs (four motifs), while NcFBP4 and NcFBP5 contained all 10 motifs. 

In order to further explore the functional characteristics of the FBP family, the phylogenetic analysis of *AtFBPs* and *NcFBPs* was carried out by using the “Find Best Homology” function of TBtools (Figure 2). The results showed that subfamily IV may be chloroplast type *FBP* (*cpFBP*) and subfamily III may be cytoplasmic *FBP* (*cyFBP*). Since another newly discovered chloroplast type *FBP* (*FBP2*) has not been confirmed in *A. thaliana* [28], it is speculated that subfamily I and subfamily II in the *NcFBPs* family belongs to *FBP2*.

### 3.3. Analysis of Cis-Acting Elements of FBP Gene Family Promoter in N. cadamba 

To assess the transcriptional regulation and potential functions of the *NcFBP* gene family, PlantCARE website was used to predict the *cis-acting* elements of the promoters of *NcFBP* gene family (Figure 3). It was found that the promoter sequences of six members of *NcFBP* gene family contained three kinds of stress-related *cis-acting* elements (drought-inducibility elements, low-temperature responsive elements and anaerobic induction elements), four kinds of hormone-related *cis-acting* elements (auxin-responsive elements, gibberellin-responsive elements, abscisic acid responsiveness elements, and MeJA-responsive elements), and four other kinds of *cis-acting* elements (circadian control elements, endosperm expression elements, light responsive elements and meristem expression elements).

The promoter region of each *NcFBP* gene contained light responsive *cis-acting* elements, which is essential for the photosynthesis. Except *NcFBP5*, other family members all contained stress-related *cis-acting* elements. These *cis-acting* elements would respond to changes in the external environment, thus regulating the expression of *NcFBP* genes, suggesting that *NcFBP* genes are closely related to the growth and development of *N. cadamba*, biotic and abiotic stress and other biological processes, and their transcriptional regulations are related to a variety of factors.

### 3.4. Analysis of Tissue Expression Pattern of NcFBPs Gene Family in N. cadamba

In order to understand the tissue expression characteristics of *FBP* genes in *N. cadamba*, the expression was analyzed by qRT-PCR (Figure 4). The results showed that the expression patterns of *NcFBP1, NcFBP4* and *NcFBP5* were similar, with high expression level in green tissues such as fruits, buds, old leaves and young leaves, while very low expression level in non-green tissues including cambium, phloem and bark. The difference of expression level between *NcFBP1* and *NcFBP4*/*NcFBP5* was that the *NcFBP1* had a certain amount of expression in the cambium, while the expression of *NcFBP4* and *NcFBP5* in non-green tissues barely occurred. The expression patterns of *NcFBP2* and *NcFBP3* were similar, with a low expression level in all tissues. The expression of *NcFBP6* was relatively stable in all tissues, however, it showed a higher expression level in tissues with relatively high metabolic rates, such as leaves, fruits, buds and cambium. Of all *NcFBP* members, the expression level of *NcFBP1* was the highest in the green tissues such as fruits, buds, old leaves and tender leaves. 

### 3.5. Phylogenetic Analysis of FBP 

A total of 157 FBPs were identified from 23 species (Table 1) including algae (*C. reinhardtii*), moss (*P. patens*), ferns (*Selaginella moellendorffii*), monocotyledons (*Setaria italica*, *Oryza sativa*, *Zea mays* PH207, *Brachypodium hybridum*, *Miscanthus sinensis*, *Sorghum bicolor*) and dicotyledons (*Citrus sinensis*, *A. thaliana*, *Populus trichocarpa*, *Amborella trichopoda*, *Eucalyptus grandis*, *Aquilegia coerulea*, *Solanum tuberosum*, *Olea europaea*, *Coffea arabica*, *Vitis vinifera*, *Solanum lycopersicum*, *Gossypium hirsutum*, *Cinnamomum kanehirae* and *N. cadamba*). The results showed that all species have less than 15 members of FBP family, and only six species (*P. patens, B. hybridum, P. trichocarpa, O. europaea, C. arabica, G. hirsutum*) have more than or equal to 10 members of FBP family, while the vast majority of species (15 of 23) have 4~7 FBP family members. The member of the FBP family of *C. reinhardtii* was the least, with only three members, while *G. hirsutum* has the most members of the FBP family, as many as 14 members. 

Figure 5 showed phylogenetic relationships and evolutionary history of FBP family in 23 species. The result shows that 157 FBPs were divided into four groups: subfamily I (28 FBPs), subfamily II (33 FBPs), subfamily III (45 FBPs) and subfamily IV (51 FBP). Except for *C. reinhardtii*, at least one member of each subfamily comes from 22 species. In each subfamily, the FBP of monocotyledons were basically clustered together, and dicotyledons had the similar pattern. In all subfamilies, the distance between *N. cadamba* and *C. arabica* was the closest, this was consistent with the results of plant taxonomy in which both *N. cadamba* and *C. arabica* were classified into Rubiaceae. 

To evaluate the patterns of evolutionary and expansion history of these gene families, we broke down the phylogenetic tree into ancestral units and estimated the most recent common ancestor (MRCA) among the 23 species (Figure 5). The results showed that there were only three MRCA shared by the 23 species, and in each subfamily, monocotyledons had specific MRCA, while dicotyledons had no specific MRCA. 

## 4. Discussion

The FBPase gene originated in bacteria in conjunction with the endosymbiotic event giving rise to mitochondria, and widely distributed in bacteria, fungi, higher plants and animals [29]. *FBP genes* have been identified and analyzed in many plants, such as potato [30,31], wheat [32,33], pea [34,35,36], spinach [37], tomato [38], *A. thaliana* [39], *Porteresia coarctata* [40], *P. haitanensis* [17], *Euglena gracilis* [41], *P. massoniana* [18] and others. To date, the genome-wide analysis of the FBP gene family has not been reported in forestry trees. In the present study, we systematically predicted and identified six *FBP* genes in the whole *N. cadamba* genome. The proteins coded by the six *FBP* gene members were negatively charged and contain FBP domain. Based on the analysis of motif, gene structure and phylogenetic tree, the six members can be divided into four subfamilies, which is consistent with the results of Li [28]. The gene structure of subfamily III was basically consistent with that of *AtcpFBP1* in *A. thaliana* [12] and *cpFBP1* in *Gossypium* species [42], and the gene structure of subfamily IV was consistent with that of *AtcyFBP* in *A. thaliana* [12] and *cyFBP* in *Gossypium* species [42], indicating that subfamily III was chloroplast type 1 FBP (cpFBP1), subfamily IV was cytoplasmic FBP (cyFBP), and both of *cpFBP1* and *cyFBP* were highly conserved in plant species, respectively. In addition, all NcFBPs have three common FBP motifs (motifs 2, 3, 4). However, motif 8 and/or motif 9 only appear in subfamily IV, which was similar to that in cotton [42]. It is speculated that subfamily IV has a special function compared with other subfamilies. Moreover, within subfamily IV, not all members have motif 8, suggesting that these motifs may be involved in the functional differentiation of subfamily IV members. 

*FBP* gene has been found to play a key role in plant photosynthesis, sugar accumulation, vascular bundle development and other growth processes in tobacco [15], potato [16], *A. thaliana* [12] and other plant species. Our study found that the most common *cis-acting* elements in the promoter region of *NcFBPs* were light responsive elements, which was similar to that in cotton [42], indicating that *FBP* gene mainly affects plant growth process by responding to light. Other studies have shown that *FBP* gene plays an important role in biotic stress [18] and abiotic stress [17,42]. This study also found that there were many *cis-acting* elements in the promoter region of *NcFBPs* related to biotic and abiotic stresses such as drought-inducibility elements, low-temperature responsive elements and anaerobic induction elements, which adds more evidence for the involvement of *FBP* gene in the process of biotic and abiotic stress. For breeders, it is necessary to further elucidate more gene functions involved in biotic and abiotic stresses, such as the ERECTA gene family [43] and DREB gene family [44]. Interestingly, a *cis-acting* element related to circadian control was found in the promoter region of *NcFBP5*, indicating that *NcFBP5* may be involved in the regulation of circadian rhythm of *N. cadamba*. This is only a preliminary prediction, and thus more research is needed to verify it. So far, it has not been reported that FBP gene was involved in the regulation of circadian control in other plant species. 

qRT-PCR was employed to analyze the tissue expression characteristics of *FBP* genes in *N. cadamba*. The expression level of each *NcFBP* gene were analyzed in different tissues, including barks, cambiums, phloems, young leaves, old leaves, fruits, and buds. The results showed that all members of *NcFBP* gene family were expressed at different levels in green tissues (young leaves, old leaves, fruits and buds), while *NcFBP1/NcFBP2/NcFBP6* were expressed at a lower level in non-green tissues. This was similar with the expression pattern of *PmFBPs* in *P. massoniana* [18], while different from the expression pattern of *HbcpFBPs* in *Hevea brasiliensis*, with a higher expression level in bark than that in leaves [45]. The reason for the different tissue expression patterns of FBP gene may be caused by different plant species, as genomic differences are associated with differences in ecological adaptation [46]. The expression patterns of *NcFBP4* and *NcFBP5* were similar, except that the expression of *NcFBP5* in green tissue was higher (Figure 4), indicating that *NcFBP5* may be chloroplast FBP, which was consistent with the previous evolutionary analysis (Figure 5). The expression of *NcFBP6* was relatively stable in all tissues, except it had a higher expression in tissues with relatively high metabolic rates (Figure 4), indicating that *NcFBP6* may be a cytoplasmic *FBP*, which was consistent with the results of previous evolutionary analysis (Figure 5). In green tissues, the gene with the highest expression was *NcFBP1*, indicating that this member may play a very important role in the green tissues. In non-green tissues such as phloem, cambium and bark, the gene with the highest expression was *NcFBP6*; this may be due to the speculation that the *NcFBP6* is a cytoplasmic *FBP* and participates in the pentosephosphate pathway, which is more important than the functions performed by other members of the FBP family in non-green organs. In future research, innovative research tools such as genome-environment associations (GEAs) [47,48,49] should be used to analyze the function and regulation mechanism of *NcFBP* genes, which can speed up the breeding process of *N. cadamba*.

The FBPs of *C. reinhardtii* does not appear in subfamily III, indicating that FBP may not differentiate until the emergence of *C. reinhardtii*. In the previous literature, cpFBPase2 (subfamily I) was not a newly evolved enzyme limited in terrestrial plants, but in the early stages of the evolution of photosynthetic organisms [28], probably in the common ancestor of photosynthetic eukaryotes. CyFBPase (subfamily II) may first replicate to produce cpFBPase2, and subsequently replicate to produce cpFBPase1 (subfamily III). The universal coexistence of these two kinds of cpFBPase in chloroplasts was likely to be the result of adapting to different redox conditions of photosynthesis, especially caused by repeated changes in light conditions. There were 12 FBP members in *P. patens*, which was four-times that in *C. reinhardtii*, indicating that the FBP family may have expanded during the evolution of algae to mosses and differentiated cpFBPase1 proteins in mosses. The FBPs of dicotyledons and monocotyledons were clustered separately by themselves, while the FBPs of algae, mosses and ferns were clustered together, which was consistent with the evolutionary status of plants. The molecular evolution mechanism of tree species is very important for forest tree breeding, while the molecular evolution mechanism of tree species is closely related to climate change. In future research, innovative methods such as predictive breeding platform [50], which can capture and harness natural tree pre-adaptations to biotic stresses by merging tools from the ecology, phylo-geography and omnigenetics fields, and modern strategies such as genomic prediction and machine learning [51] are needed to assess and breed forest tree adaptation to the changing climate [52]. This may provide technical support for tree breeding projects to achieve the purpose of accelerating breeding.

## 5. Conclusions

In this study, a total of six *FBP* genes were identified and cloned based on the *N. cadamba* genome, and comprehensively analyzed in their molecular evolution including phylogenetic relationships, gene structures and conserved motifs. The six *NcFBP* genes were divided into four groups according to phylogenetic analysis. This study provide evidence that there are many *cis-acting* elements related to biotic and abiotic stresses in the promoter region of *NcFBPs*, and a *cis-acting* element related to circadian control first found in the promoter region of FBP gene. Furthermore, in combination with phylogenetic analysis, analyses of the expression profiles of six *NcFBP* genes based on qRT-PCR in various *N. cadamba* tissues provide evidence that *NcFBP5* may be chloroplast type 1 *FBP*, and *NcFBP6* may be cytoplasmic *FBP*. The phylogenetic relationships and evolutionary history of FBP family in 23 species showed that FBP family may have expanded during the evolution of algae to mosses and differentiated cpFBPase1 proteins in mosses. This study represents the first investigation of the FBP gene family in *N. cadamba*, providing a foundation for further study on the function of the FBP genes in this species.

## Figures and Tables

**Figure 1 genes-13-02349-f001:**
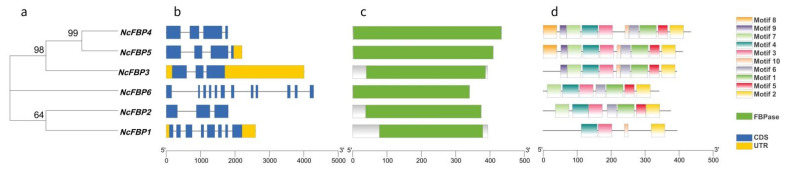
Phylogenetic relationships, gene structure, domains and motifs of FBP family in *N. cadamba*. (**a**) Protein maximum likelihood (ML) tree. (**b**) Gene structure, introns were shown as lines, exons and UTR were shown as blue and yellow rectangle, respectively. (**c**) The conserved domains, FBPase domains (PF00316) were highlighted by green. (**d**) The conserved motifs, all motifs were identified by MEME database.

**Figure 2 genes-13-02349-f002:**
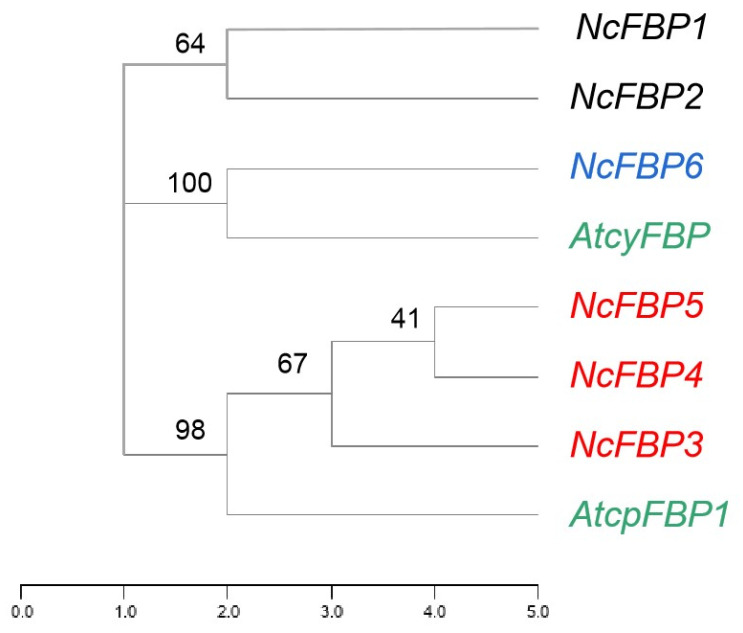
Evolutionary Analysis of *FBP* homologous genes between *N. cadamba* and *A. thaliana*. The *AtcyFBP* and *AtcyFBP1* represent the cytoplasmic *FBP* and the chloroplast *FBP1* of *A. thaliana*, respectively.

**Figure 3 genes-13-02349-f003:**
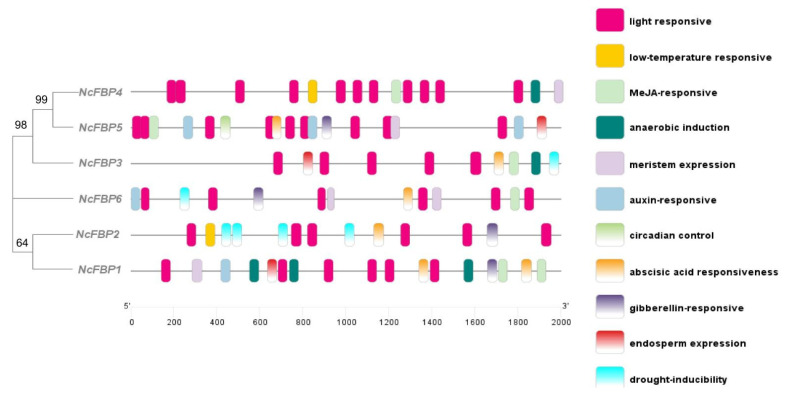
Prediction of cis-acting elements in the promoter regions of *FBP* gene family in *N. cadamba*.

**Figure 4 genes-13-02349-f004:**
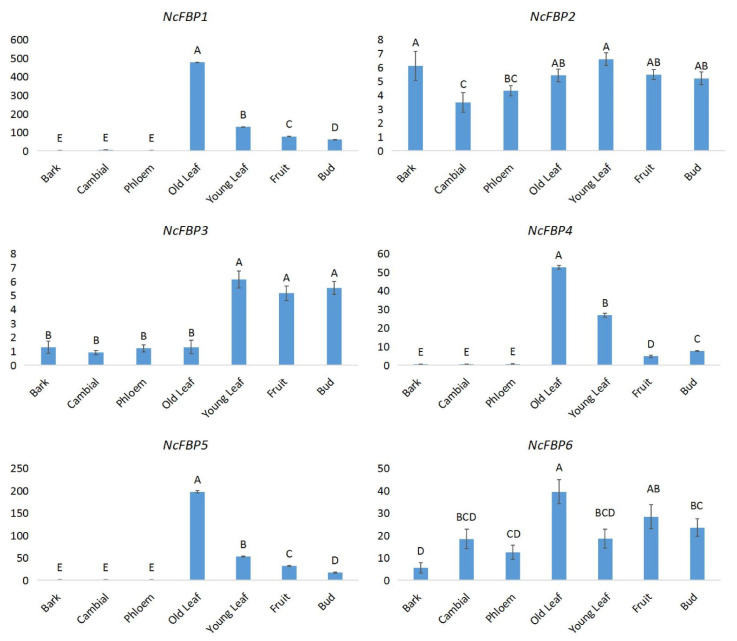
The expression patterns of *NcFBPs* genes in tissues of *N. cadamba*. The same capital letters indicate groups that are not significantly different from each other according to Duncan’s multiple range test, *p* = 0.01.

**Figure 5 genes-13-02349-f005:**
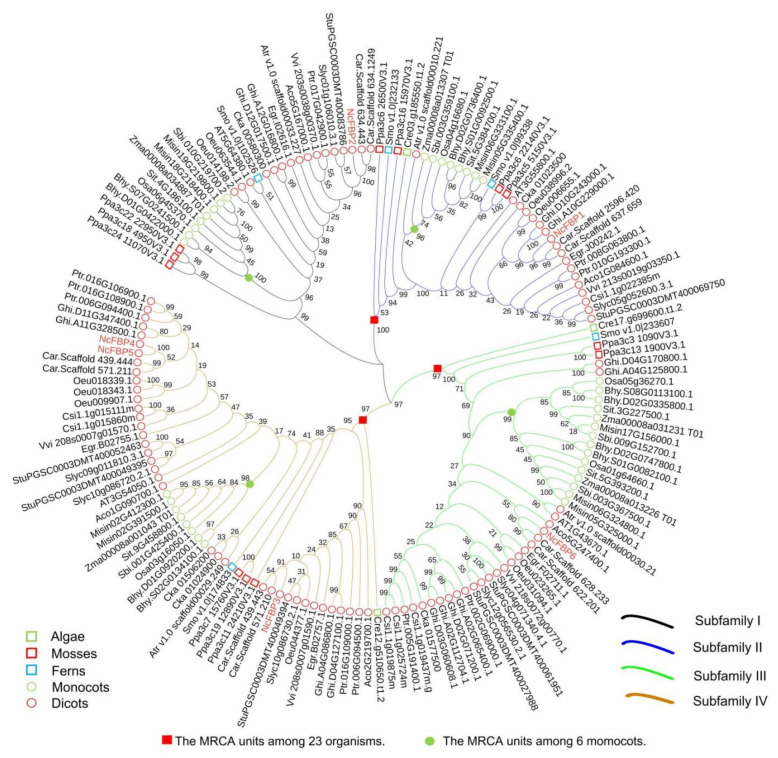
Phylogenetic relationships and evolutionary history of FBP family in 23 species. Cre represents *C. reinhardtii*, Ppa represents *P. patens*, Smo represents *S. moellendorffii*, Sit represents *S. italica*, Osa represents *O. sativa*, Zma represents *Z. mays*, Bhy represents *B. hybridum*, Msi represents *M. sinensis*, Sbi represents *S. bicolor*, Csi represents *C. sinensis*, Ath represents *A. thaliana*, Ptr represents *P. trichocarpa*, Atr represents *A. trichopoda*, Egr represents *E. grandis*, Aco represents *A. coerulea*, Stu represents *S. tuberosum*, Oeu represents *O. uropaea*, Car represents *C. arabica*, Vvi represents *V. vinifera*, Sly represents *S. lycopersicum*, Ghi represents *G. hirsutum*, Cka represents *C. kanehirae*, Nc represents *N. cadamba*.

**Table 1 genes-13-02349-t001:** Genome-wide identification of FBP in the 23 completely sequenced plant genomes.

Classification	Species Name	Common Name	Genome Size (Mb)	Org Code	FBP (PF00316)
Algae	*Chlamydomonas reinhardtii*	Green algae	111.1	Cre	3
Mosses	*Physcomitrella patens*	Moss	473.0	Ppa	12
Ferns	*Selaginella moellendorffii*	Spikemoss	212.5	Smo	5
Monocots	*Setaria italica*	Foxtail millet	405.7	Sit	5
Monocots	*Oryza sativa*	Rice	372.0	Osa	5
Monocots	*Zea mays PH207*	Maize	2450.0	Zma	5
Monocots	*Brachypodium hybridum*		509.0	Bhy	10
Monocots	*Miscanthus sinensis*	Chinese silver grass	2000.0	Msi	9
Monocots	*Sorghum bicolor*	Cereal grass	732.2	Sbi	5
Dicots	*Citrus sinensis*	Sweet orange	319.0	Csi	6
Dicots	*A. thaliana*	Thale cress	135.0	Ath	4
Dicots	*Populus trichocarpa*	Western poplar	422.9	Ptr	11
Dicots	*Amborella trichopoda*	Amborella	706.0	Atr	4
Dicots	*Eucalyptus grandis*	Eucalyptus	691.0	Egr	5
Dicots	*Aquilegia coerulea*	Colorado blue columbine	306.5	Aco	5
Dicots	*Solanum tuberosum*	Potato	723.0	Stu	7
Dicots	*Olea europaea*	Olive	1140.0	Oeu	10
Dicots	*Coffea arabica*	Coffee	1192.6	Car	10
Dicots	*Vitis vinifera*	Grape vine	487.0	Vvi	5
Dicots	*Solanum lycopersicum*	Tomato	900.0	Sly	7
Dicots	*Gossypium hirsutum*	Upland cotton	2305.2	Ghi	14
Dicots	*Cinnamomum kanehirae*	Stout camphor tree	730.4	Cka	5
Dicots	*N. cadamba*	Kadan	724.5	Nca	6

**Table 2 genes-13-02349-t002:** Physicochemical properties of FBP protein in *N. cadamba*.

Gene Symbol	Gene ID	pI	MW (kDa)	aa-num	Instability Index	Aliphatic Index	Grand Average of Hydropathicity
*NcFBP1*	evm.model.Contig96.207	5.76	42.5726	393	35.88	86.11	−0.092
*NcFBP2*	evm.model.Contig54.150	5.77	40.4778	374	37.52	94.04	−0.108
*NcFBP3*	evm.model.Contig462.135	6.27	43.0109	392	44.59	89.29	−0.223
*NcFBP4*	evm.model.Contig462.133_evm.modle.Contig462.134	5.45	44.3803	407	42.64	88.38	−0.156
*NcFBP5*	evm.model.Contig296.294	5.25	44.3753	409	37.70	91.27	−0.120
*NcFBP6*	evm.model.Contig172.7	5.87	37.1957	340	36.53	84.56	−0.168

## Data Availability

Not applicable.

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
