# Peer review of "Evolution and Expression Patterns of the Fructose 1,6-Bisphosptase Gene Family in a Miracle Tree (Neolamarckia cadamba)"

_genes, 2022, doi:10.3390/genes13122349_

Round 1
Reviewer 1 Report
The work by Que et al. explored the genetic diversity, genealogies and expression patterns of six fructose 1,6-bisphosptase (FBP) genes in Neolamarckia cadamba, a promising tree species in South East easy due to its fast growth. This is an accurate and pertinent work, which addresses the genetic and expression diversity in a non-model tree species. Overall, the work is well written, statistically up to date, and highlights key findings.
I have the following suggestions.
First, the paragraphs of the introduction must be reverted so that authors first describe the importance of the tree (paragraph in L66 must go first) and later on start explaining the peculiarities of the FBP gene family. Please follow this structure for the abstract as well.
Second, I recognize authors for mentioning in L47 and L318 the broad consequences of FBP genes for abiotic stresses responses, and the specificities for drought responses in L60, L220 and L320. Still, I am missing key references regarding the polygenetic nature of the genomic architecture for abiotic stresses that may be pleiotropic with more complex effects beyond the FBP gene family. Specifically, other gene families have also been linked with abiotic stresses, drought in particular, as indicated in: (i) Plant Science 2016 242:250 for the ERECTA gene family in association with the AP2 domain, (ii) BMC Genetics 2012 13:58 for the ASR family f in association with the ABA-dependent MYB, and (iii) Theor Appl Genet 2012 125(5):1069-85 for the DREB transcription factor family pleotropic in several pathways with the WRKY transcription factor. Authors should discussed these cases explicitly and make a clear point since the introduction on why focusing only on the FBP gene family seems biased and these points in the introduction and discussion sections. Please revisit this point at the end of the discussion and recommend expanding the analysis to other gene families.
Third, my major analytical suggestion is to leave out the supplemental tables and do not include them in the main version of this manuscript (L397). The primer details exceed the need of the main manuscript. Besides, every time a dendogram is presented (figures 1-5) please include significance as bootstrap values on top of the branches.
Fourth, please also complement the tissue-dependent expression in figure 4 with explicit gene-environment associations, which may have higher power for the detection of context-dependent environmental responses (refer to the seminal review Front Genet 2022 13:910386). The gene-environment approach has been validated for heat (refer to Front Genet 2019 10:954) and drought (refer to Front Plant Sci 2018 9:128 and Genes 2021 12:556), stresses, some additional resources authors may refer to. Also, authors should explicitly comments on the significance of the replication level utilized to gather the expression profiles, for which a preliminary power analysis would be insightful.
As closure, please include a perspective section at the end of the introduction in L363 with recommendation on how to better integrate omic technologies with modern analytical approaches to assess adaptation and abiotic stress tolerance in tree species by referring to the seminal reviews Front Plant Sci 2020 11:583323 and Front Genet 2020 11:564515, and its link with other type of stresses in forest trees (e.g. biotic pressures as discussed in Plants 2021 10:2022). These recommendations would be useful for readers considering planting N. cadamba in other regions of the world outside its native distribution in South East Asia. For sure future studies on adaptation in miracle trees would benefit from these fresh innovative perspectives.
Finally, in terms of writing, the abstract is excessively synthetic and does not follow the ABT recommendation (see this card: https://entomologychallenges.files.wordpress.com/2018/10/abt-shorthand-reference-card.pdf) for abstracts (in L13). Please improve it.
Related with the previous point, please enlist explicit research gap, research hypotheses and goals at the end of the introduction (L81). This will allow readers focusing on explicit expectations when approaching the report.
Author Response
Comment 1: First, the paragraphs of the introduction must be reverted so that authors first describe the importance of the tree (paragraph in L66 must go first) and later on start explaining the peculiarities of the FBP gene family. Please follow this structure for the abstract as well.
Response1:Thanks for the reviewer’s suggestion. We have moved the describe of the tree to first paragraph, and the abstract has also been revised.
Comment 2: Second, I recognize authors for mentioning in L47 and L318 the broad consequences of FBP genes for abiotic stresses responses, and the specificities for drought responses in L60, L220 and L320. Still, I am missing key references regarding the polygenetic nature of the genomic architecture for abiotic stresses that may be pleiotropic with more complex effects beyond the FBP gene family. Specifically, other gene families have also been linked with abiotic stresses, drought in particular, as indicated in: (i) Plant Science 2016 242:250 for the ERECTA gene family in association with the AP2 domain, (ii) BMC Genetics 2012 13:58 for the ASR family f in association with the ABA-dependent MYB, and (iii) Theor Appl Genet 2012 125(5):1069-85 for the DREB transcription factor family pleotropic in several pathways with the WRKY transcription factor. Authors should discussed these cases explicitly and make a clear point since the introduction on why focusing only on the FBP gene family seems biased and these points in the introduction and discussion sections. Please revisit this point at the end of the discussion and recommend expanding the analysis to other gene families.
Response2: Thanks for the reviewer’s suggestion. We have expanded the analysis to other gene families: For breeders, it is necessary to clarify more about the gene functions of biotic and abiotic stresses, such as the ERECTA gene family[43], DREB gene family[44].
Comment 3: Third, my major analytical suggestion is to leave out the supplemental tables and do not include them in the main version of this manuscript (L397). The primer details exceed the need of the main manuscript. Besides, every time a dendogram is presented (figures 1-5) please include significance as bootstrap values on top of the branches.
Response3: Thanks for the reviewer’s suggestion. We have moved out the supplemental tables, and added bootstrap values on top of the branches.
Comment 4: Fourth, please also complement the tissue-dependent expression in figure 4 with explicit gene-environment associations, which may have higher power for the detection of context-dependent environmental responses (refer to the seminal review Front Genet 2022 13:910386). The gene-environment approach has been validated for heat (refer to Front Genet 2019 10:954) and drought (refer to Front Plant Sci 2018 9:128 and Genes 2021 12:556), stresses, some additional resources authors may refer to. Also, authors should explicitly comments on the significance of the replication level utilized to gather the expression profiles, for which a preliminary power analysis would be insightful.
Response4: Thanks for the reviewer’s suggestion. We have replaced the original heatmap (figure 4) with bar graphs, and analyzed the expression pattern of NcFBP genes in different tissues using Duncan's multiple comparisons. Furthermore, we included knowledge related to GEA in the discussion: In the future research, innovative research tools such as genome-environment associations (GEAs)[47,48] should be used to analyze the function and regulation mechanism of NcFBP genes, which can speed up the breeding process of N.cadamba.
Figure 4 The expression patterns of NcFBPs genes in tissues of N.cadamba
Note: The same capital letters indicate groups that are not significantly different from each other according to Duncan’s multiple range test, p = 0.01.
Comment5: As closure, please include a perspective section at the end of the introduction in L363 with recommendation on how to better integrate omic technologies with modern analytical approaches to assess adaptation and abiotic stress tolerance in tree species by referring to the seminal reviews Front Plant Sci 2020 11:583323 and Front Genet 2020 11:564515, and its link with other type of stresses in forest trees (e.g. biotic pressures as discussed in Plants 2021 10:2022). These recommendations would be useful for readers considering planting N. cadamba in other regions of the world outside its native distribution in South East Asia. For sure future studies on adaptation in miracle trees would benefit from these fresh innovative perspectives.
Response5: Thanks for the reviewer’s suggestion. We have revised and added some innovative methods such predictive breeding platform the discussion: The molecular evolution mechanism of tree species is very important for forest tree breeding,while molecular evolution mechanism of tree species is closely related to climate change. In future research, innovative methods such as predictive breeding platform[49], which can capture and harness natural tree pre-adaptations to biotic stresses by merging tools from the ecology, phylo-geography, and omnigenetics fields, and modern strategies are needed to assess and breed forest tree adaptation to changing climate[50]. That may provide technical support for tree breeding project to achieve the purpose of accelerating breeding..
Comment 6: Finally, in terms of writing, the abstract is excessively synthetic and does not follow the ABT recommendation (see this card: https://entomologychallenges.files.wordpress.com/2018/10/abt-shorthand-reference-card.pdf) for abstracts (in L13). Please improve it.
Response6: Thanks for the reviewer’s suggestion. We have revised the abstract followed by the ABT recommendation: Neolamarckia cadamba is a fast-growing tree species with tremendous economic and ecological value, the study of the key genes regulating photosynthesis and sugar accumulation is very important for the breeding of N.cadamba. Fructose 1,6-bisphosptase (FBP) gene has been found to play a key role in plant photosynthesis, sugar accumulation, and other growth processes. However, no systemic analysis of FBPs has been reported in N.cadamba. A total of 6 FBP genes were identifed and cloned based on the N.cadamba genome, and these FBP genes were sorted into 4 groups. The characteristics of the NcFBP gene family were analyzed such as phylogenetic relationships, gene structures, conserved motifs, and expression patterns. A cis-acting element related to circadian control was first found in the promoter region of FBP gene. Phylogenetic and quantitative real-time PCR analyses showed that NcFBP5 and NcFBP6 may be chloroplast type 1 FBP and cytoplasmic FBP, respectively. FBP proteins from N.cadamba and 22 other plant species were used for phylogenetic analyses, indicating that FBP family may have expanded during the evolution of algae to mosses and differentiated cpFBPase1 proteins in mosses. This work analyzes the internal relationship between the evolution and expression of the 6 NcFBPs, providing a scientific basis for the evolutionary pattern of plant FBPs, and promoting the functional studies of FBP genes.
Comment 7: Related with the previous point, please enlist explicit research gap, research hypotheses and goals at the end of the introduction (L81). This will allow readers focusing on explicit expectations when approaching the report.
Response7:Thanks for the reviewer’s suggestion. We have added research gap, research hypotheses and goals at the end of the introduction: As FBPase plays a key role in plant photosynthesis, sugar accumulation, and other growth processes, the study of FBPase is important for N. cadamba breeding. In the study, the NcFBP gene family members were cloned in the N. cadamba genome, followed by comprehensive analyses, including sequence alignment, gene structures, phylogenetic analyses, conserved motifs, and conserved domains. Theremore, the expression patterns of FBP genes in different tissues in N. cadamba and their phylogenetic analysis were investigated.

Reviewer 2 Report
Title. The title conveys the main message of the paper — the issues addressed and the relationships among the issues.
Abstract. The abstract is concise, provides a clear overview, includes essential facts for the paper, and concludes with a final point that places the work described in a broader context.
Keywords. These are enough for the topic.
Introduction. The introduction includes background to provide an appreciation for the context of the work presented and also states the rationale and problem that the researchers attempted to answer through their experiments.
Material and methods. In this section, the authors describe the correct steps that followed during conducting their study, give precise details of the study design, and how they analyzed the data.
Results and discussion. These sections were well written and shows all data with good descriptions. The results say about the objective that motivates the research, and the authors take a broad look at their findings and examine the work in the larger context of the field. The authors had to discuss the data with respect to how their data fit into what is currently known in the field.
Conclusion. This section included the major conclusions, which were briefly written.
Figures and Tables. Both sections have good information and are necessary for the manuscript, they depict the data nicely.
Author Response
Comment 1:
Title. The title conveys the main message of the paper — the issues addressed and the relationships among the issues.
Abstract. The abstract is concise, provides a clear overview, includes essential facts for the paper, and concludes with a final point that places the work described in a broader context.
Keywords. These are enough for the topic.
Introduction. The introduction includes background to provide an appreciation for the context of the work presented and also states the rationale and problem that the researchers attempted to answer through their experiments.
Material and methods. In this section, the authors describe the correct steps that followed during conducting their study, give precise details of the study design, and how they analyzed the data.
Results and discussion. These sections were well written and shows all data with good descriptions. The results say about the objective that motivates the research, and the authors take a broad look at their findings and examine the work in the larger context of the field. The authors had to discuss the data with respect to how their data fit into what is currently known in the field.
Conclusion. This section included the major conclusions, which were briefly written.
Figures and Tables. Both sections have good information and are necessary for the manuscript, they depict the data nicely.
Response1:Thanks for the reviewer’s suggestion. We have further refined the manuscript.

Round 2
Reviewer 1 Report
Unfortunately authors have failed to include highly relevant reviews/citations (some of them suggested in the previous revision, please check it in detail). They are desired in order to better give context to the work and the methodological framework. Please carefully consider their inclusion and discussion
Author Response
Comment 1: Unfortunately authors have failed to include highly relevant reviews/citations (some of them suggested in the previous revision, please check it in detail). They are desired in order to better give context to the work and the methodological framework. Please carefully consider their in conclusion and discussion.
Response1:Thanks for the reviewer’s suggestion. We have added all the references mentioned in the previous review comments to the citation and discussion, except for ‘BMC Genetics 2012 13:58’, which does not fit very well with the topic of our article. Thanks again for the reviewer’s comments and suggestions, that make our article more meaningful.